# Synthesis, Characterisation, Biological Evaluation and In Silico Studies of Quinoline–1,2,3-Triazole–Anilines as Potential Antitubercular and Anti-HIV Agents

**DOI:** 10.3390/molecules30102119

**Published:** 2025-05-10

**Authors:** Snethemba S. Magwaza, Darian Naidu, Oluwatoba E. Oyeneyin, Sibusiso Senzani, Nompumelelo P. Mkhwanazi, Matshawandile Tukulula

**Affiliations:** 1School of Chemistry and Physics, University of KwaZulu Natal, Westville Campus, Durban 4001, South Africa; magwazas@ukzn.ac.za; 2HIV Pathogenesis Programme, Doris Duke Medical Research Institute, School of Laboratory Medicine and Medical Sciences, College of Health Sciences, University of KwaZulu Natal, Durban 4001, South Africa; 218052764@stu.ukzn.ac.za (D.N.); mkhwanazi@ukzn.ac.za (N.P.M.); 3Department of Chemical Sciences, Adekunle Ajasin University, Akungba-Akoko 342111, Nigeria; oluwatoba.oyeneyin@aaua.edu.ng; 4School of Laboratory Medicine and Medical Sciences, College of Health Sciences, University of KwaZulu Natal, Durban 4001, South Africa; senzanis@ukzn.ac.za

**Keywords:** quinoline–1,2,3-triazole–anilines, click reaction, HIV-1 subtype B, molecular docking, DFT (density  functional theory)

## Abstract

HIV/AIDS and *Mycobacterial tuberculosis* (*Mtb*) are the leading cause of deaths worldwide. Thus, better medicaments are required to manage these diseases. Quinolines have shown great potential due to their broad spectrum of biological activity. Thus, quinoline–1,2,3-triazole–aniline hybrids were synthesised in moderate to good yields. Compounds **11g** (IC_50_ = 0.388 µM), **11h** (IC_50_ = 0.01032 µM) and **11i** (IC_50_ = 0.167 µM) exhibited the most promising in vitro activities against the wild-type HIV-1 subtype B, with **11h** being 9-fold more active than AZT (IC_50_ = 0.0909 µM), the reference drug. Furthermore, compound **11h** displayed moderate activity, with a MIC_90_ of 88μM against *Mtb*’s H37Rv strain. Cytotoxicity studies on TZM-bl cell lines revealed that most of the tested compounds were generally non-cytotoxic; the selectivity index (SI) for **11h**, the front runner, is >2472. Molecular docking studies revealed that **11h** interacted with Phe112, Tyr108, Glu283 and Trp86 amino acid residues in the active site of HIV-1. DFT studies revealed that **11h** has the ability to donate and accept electrons to and from available orbitals. The predicted ADMET studies showed that these compounds possess drug-likeness, and **11h** has the potential for further optimisation as an anti-HIV-1 agent.

## 1. Introduction

Africa and other developing countries have borne a substantial burden of mortality attributed to infectious diseases [1,2]. One of the most concerning infectious diseases, tuberculosis (TB), is caused by *Mycobacterium tuberculosis* (*Mtb*), a Gram-negative bacterium with a lipid-rich outer membrane that makes it resilient against various antibiotics [3,4,5]. To treat drug-susceptible TB, combinations of isoniazid, rifampicin, pyrazinamide, and ethambutol, termed first-line drugs, are administered [6,7,8]. Each of these drugs targets specific areas in *Mtb*, with the shared objective of inhibiting the production of essential biological processes critical for bacterial replication and survival [6]. Over the years, due to gene mutations, *Mtb* has evolved to more resistant isoforms, such as multidrug-resistant (MDR-TB), extensively drug-resistant (XDR-TB) and totally drug-resistant TB (TDR-TB) [9,10].

On the other hand, the human immunodeficiency virus (HIV) remains one of the most prevalent infectious diseases [11]. To date, 39.9 million people are living with HIV/AIDS (PLWH) worldwide [12]. Effective drug development over the years has saved many lives, with the discovery of highly active antiretroviral therapy (HAART) being the pivotal moment. The use of antiretroviral drugs has decreased the morbidity and mortality rate of PLWH by turning HIV-1 into a chronic and manageable disease [13,14]. Despite HAART’s early success, drug-resistant mutant development has rendered it less effective over time. HIV is a complex virus with a high mutation rate, considering it replicates in several replication stages. Antiretroviral drugs target different stages of the HIV-1 life cycle: viral attachment, reverse transcription, integration, proteolysis and viral budding [15]. There are seven classes of antiretrovirals based on their molecular mechanisms and resistance profiles: nucleoside-analogue reverse transcriptase inhibitors (NNRTIs), non-nucleoside reverse transcriptase inhibitors (NNRTIs), integrase inhibitors, protease inhibitors (PIs), fusion inhibitors, post-attachment inhibitors (PAIs) and co-receptor antagonists [16,17,18,19]. These drug regimens constantly evolve, and new drugs are continually being developed in each class.

Various quinoline- and/or 1,2,3-triazole-containing compounds have been reported to show promising activities against either *Mtb* [20,21,22,23,24,25,26,27,28] or HIV infections [29,30,31,32], with some already in clinical settings, including the 1,2,3-triazole-pyrimidine hybrids that are now part of the second-generation NNRTIs [30]. Bedaquiline (or TMC-207) (Figure 1), a quinoline-based compound, is clinically utilised for treating multidrug-resistant TB [33]. This compound binds to the subunit C of mycobacterial ATP synthase, an essential enzyme for *Mtb* energy production and survival. However, bedaquiline suffers from several side effects due to its potent inhibition of the potassium ether-ago-go-related gene (hERG) that could potentially lead to cardiac arrest [34]. Thus, new quinoline-based anti-*Mtb* agents, devoid of bedaquiline’s shortcomings, are critically important. Due to their broad spectrum of activities, various quinoline–triazole hybrids have shown promise over the years. Thomas et al. [21] reported on a series of new 6-methoxyquinoline triazole amides (**1**) (MIC = 0.625 µg/mL), sulphonamides (**2**) (MIC = 0.625 µg/mL) and amidopiperazines (**3**) (MIC = 0.625 µg/mL) that exhibited promising antitubercular activities. Our group recently disclosed a series of 7-chloroquinoline–triazole–benzimidazole hybrids that demonstrated excellent *Mtb* activity, with isomeric mixture **4** showing a MIC_90_ of 1.49 µM [22]. Previously, Costa et al. [29] reported several quinoline–1,2,3-triazole hybrids, such as **5** (IC_50_ = 800 nm), which showed promising activity against HIV reverse transcriptase. Maraviroc, the first licensed CCR5 co-receptor antagonist containing the triazole moiety, is used to treat HIV infections and is less prone to drug resistance than the presently used ([N]NRTI) [35,36,37].

Simultaneously targeting TB and HIV is of paramount importance, especially considering the high prevalence of co-infection, drug resistance and the potential for drug–drug interactions in patients receiving treatment for both diseases [38]. Combining quinoline and triazole scaffolds into hybrid structures may synergistically improve the efficacy against both *Mtb* and HIV [39,40]. The structural analysis of some of the compounds shown in Figure 1 reveals common structural features, shown in blue in Figure 2. Could these structural features be responsible for these compounds’ combined anti-*Mtb* and anti-HIV activities? Thus, in this study, we report on the synthesis, biological evaluation and computational studies of a series of compounds containing these structural features, albeit with the substitution of the quinoline moiety limited to the 7-chloro only.

## 2. Results and Discussion

### 2.1. Chemistry

The target compounds were synthesised in two stages, as shown in Figure 1. The first stage was the formation of the two key intermediates, namely the 7-chloroquinoline-4-azide **7** [22] and the alkyne **10a**–**j** [41,42,43,44], via steps 1 and 2 according to the reported literature procedures and their spectroscopic data confirmed from these reports also. These intermediates were then subjected to “click chemistry” [27] in step 3 to yield the final quinoline–1,2,3, triazole–aniline derivatives, **11a**–**j**, in 43 to 92% yields (Table 1). Then, 1D and 2D nuclear magnetic resonance (NMR) and infrared (IR) spectroscopy were used to confirm the structures, while mass spectrometry confirmed the masses of the desired products (see Appendix A).

Confirmation of the formation of 1,2,3-triazole was achieved by observing the chemical shift of the singlet triazole methine proton, which appears in the aromatic region around δ_H_ 8.7 ppm. Secondly, the methylene doublet, due to coupling with the neighbouring NH, appeared as expected at around 4.47 ppm, while the NH appeared as a triplet at 6.21 ppm. All the quinoline protons appeared as expected. The ^13^C NMR ATP NMR analysis also confirmed the formation of the hybrid, with the critical methine and methylene resonances appearing at 125.68 and 38.95 ppm, respectively.

The heteronuclear multiple bond correlation (HMBC) experiment (Figure 3) further validated the triazole ring’s formation, with close correlations of H6-C13, H1-C13, H1-C6 and H6-C14 being observed and shown in the structure of **11a**. The mass spectrometry confirmed the masses of the target compounds based on their main or major fragment as observed in the base peak. All the other compounds were characterized in a similar manner.

### 2.2. In Vitro Biological Activities

The synthesised quinoline–1,2,3-triazole–anilines were evaluated in vitro against the *Mtb* H37Rv (ATCC 27294) strain and against HIV-1 subtype B, and their cytotoxicity was assessed using an MTT assay on the TZM-bl cell line (Table 2). The *Mtb* activity is represented as the 90% minimum inhibitory concentration (MIC_90_), while HIV and cytotoxicity are represented as the 50% inhibitory concentration (IC_50_) and 50% cytotoxicity concentration (CC_50_), respectively. In terms of anti-*Mtb*, the data reveal that all the synthesised hybrid compounds did not show any appreciable activity, with compounds **11b**, **11c**, **11i** and **11j** exhibiting poor activity (MIC_90_ > 1000 μM), while **11a**, **11d**–**g**, and **11i** fell within the range of 100–200 μM. Notably, compound **11h** exhibited the most potent activity, with a MIC_90_ of 88 μM, distinguishing it as the most active among all of these, albeit 9-fold less active than the reference drug, ethambutol. Interesting the quinoline azide intermediate (**7**) was the more active than the synthesised hybrids, while the propargylated unsubstituted aniline (**10a**) was the least active of all the tested compounds. The activities of the intermediates versus the hybrids indicated that the hybridization strategy was antagonistic with regards to the quinoline azide, while synergistic with regards to the propargylated aniline.

Against HIV-1, initially, all the compounds that showed a > 50% inhibition in the primary inhibition assay were progressed to determine their IC_50_, and azidothymidine (AZT) was used as a control drug. Generally, all the compounds inhibited the growth of the HIV-1 subtype B virus, ranging from a 58 to 100% inhibition potential (see percentage cell viability plots in the Appendix A). The IC_50_ values for most of these compounds were moderate, with only three compounds exhibiting sub-micromolar activity, namely **11g**, **11h** and **11i**, albeit not as active as AZT. Compound **11g** (consisting of the trifluoromethyl substituent on the phenyl ring) (IC_50_ = 0.3883 µM) and **11i** (the 3-nitrosubstituted) (IC_50_ = 0.170 µM) were 4- and 1.8-fold less active than AZT, respectively. On the other hand, **11h** (the 3-flourosubstituted) (IC_50_ = 0.01032 µM) was 8.8 times more potent than AZT. Interestingly, compound 11h was also the most active against *Mtb*, highlighting its dual active potential. Cytotoxicity assessment revealed that five of the synthesised compounds, **11a**, **11c**–**e** and **11g**, exhibited a CC_50_ >100 µM, with **11g** showing the best cytotoxicity profile, with a CC_50_ = 4414 µM and selectivity index (SI) of > 11,300, which is not far off from that of AZT (SI = 12,349). On the other hand, the CC_50_ value of compound **11h** was 25.52 µM and its SI > 2400, an indication of it being less likely to be cytotoxic in vivo [45,46]. Thus, compound **11h** is a potential “hit” compound for further optimisation as a potential anti-HIV agent based on its activity and cytotoxicity profile.

### 2.3. In Silico Studies

#### 2.3.1. Molecular Docking

To validate the observed biological activities, all the hybrid compounds underwent in silico docking simulations into the active site of the *Mtb* ATP synthase enzyme (PDB ID: 4VIF) [47] and the antiviral enzyme (PDB ID: 4MBS) [48] using the Maestro software 13.9 in the Schrödinger Suite [49]. The docking scores recorded in Table 3 ranged from −2.879 to −2.035 kcal/mol for the antimicrobial target and −7.371 to −4.815 kcal/mol for the antiviral target. Compound **11d** had the best docking score, followed by **11e** and **11h** against the antiviral enzyme, while **11e** had the best docking score against the antimicrobial ATP synthase, followed by **11g** and **11h**. Notably, the docking scores were lower for the TB target than for the HIV target, consistent with the obtained biological data. The two promising compounds in the anti-TB and anti-HIV biological screenings, and their interaction with the active sites of the proteins, were observed as depicted in Figure 4.

The 4V1F enzyme is known to be the target for compounds containing the quinoline moiety, such as mefloquine and bedaquiline, and it has been used extensively in the molecular docking of quinoline-based compounds [22,43]. Compound **11h** (MIC_90_ = 88.72 µM) displayed a docking score of −2.606 compared to **11a** (MIC_90_ = 186.52 µM), which had a docking score of −2.540. In Figure 4, compound **11a** demonstrates hydrogen bond interactions with Glu65, Tyr68 and Phe69 and additional interactions with Ala66, Gly62, Val61, and Phe58 amino acid residues. On the other hand, **11h** exhibited similar interactions with amino acid residues but without any visible hydrogen interaction. Other biological properties of **11h** may contribute further to its biological activity.

The selected CCR5 chemokine receptor acts as a co-receptor for HIV-1 viral entry, and its associated enzyme (4 MBS) is a known target for maraviroc, a triazole-containing antiretroviral [36]. Furthermore, Singh et al. [50] and Ibrahim et al. [51] reported quinoline-based compounds as chemokine receptor CCR5 inhibitors. Compound **11h** (IC_50_ = 0.01032 µM) demonstrated a significantly higher docking score of −7.362 than **11a** (IC_50_ = 3.013 µM). Hydrogen-bonding interactions were observed with various amino acid residues in the active sites; the quinoline and aniline moieties interacted with Phe112, Tyr108, Glu283 and Trp86 for **11h** and **11a**. Further interactions with Tyr89 and Trp248 amino acid residues were observed for both compounds.

#### 2.3.2. Density Functional Theory Studies

The stability of a compound is a consequence of its orbital energies [52]. Table 4 shows the stability and/or reactivity indices of **11h** at the DFT/B3LYP/6-311++G(d,p) level of theory. These reactivity indices are vital for molecular reactivity. The ionisation potential (I) and the electron affinity (A) of **11h** were 5.89 eV and 2.64 eV, respectively. A low I value suggests that a molecule can give up electrons readily [53]. A high value of A implies a good electron-accepting potential for the molecules [53]. The energy gap (Eg) is derived as the difference between the frontier orbital energies. The Eg of **11h** was 3.25 eV, while its chemical hardness was 1.63 eV. The global softness was 0.615 eV^−1^. The ability of a molecule to attract an electron is related to the electronegativity, χ. The χ of **11h** was 4.27 eV, while its electrophilicity was 5.59 eV. The values of the reactivity descriptors in 11h here are close to reported bioactive compounds at the same level of theory [54]. Compound **11h** can give up electrons easily and also accept electrons, from the values of its ionisation potential and electron affinity. These properties are important to biological systems in that they provide likely indications of a molecule’s ability to react chemically, even with biological systems such as proteins and/or enzymes, albeit without providing the actual activity against a particular cell line or pathogens [55,56,57].


**HOMO, LUMO and ESP surface maps of 11h**


The optimised structure, HOMO, LUMO and electrostatic potential maps of **11h** are shown in Figure 5. The HOMO map of the compound was spread across the entire fluoroaniline moiety, while the LUMO was delocalised over the other side of the compound (quinoline and triazole rings). This indicates that the compound has the ability to act as an electron donor, as well as an acceptor of electrons.

The electrostatic potential (ESP) map (Figure 6) accounts for the regions in a molecule prone to nucleophilic and electrophilic attacks. While the red- and yellow-mapped regions show a negative electrostatic potential and are prone to attack by an electrophile, the blue- and/or green-mapped regions indicate a positive electrostatic potential and are prone to nucleophilic attack.

The sites prone to electrophilic attack in **11h** are the fluorine and chlorine atoms, as well as the quinoline nitrogen. The sites prone to nucleophilic attack are the triazole’s C19, C23, N21, N22 and H30 atoms, extending to N18 through C24 and their hydrogen atoms; some faint blue/green maps were seen on other parts of the quinoline rings. The charge separation between these two opposite electrostatic potential sites could facilitate intramolecular charge transfer between the molecule and biological systems and identify potential binding sites for any substances and/or inhibitors [55,56,57,58].

#### 2.3.3. ADMET Predictions

Adsorption, distribution, metabolism, excretion and toxicity (ADMET) properties describe a drug’s absorption, distribution, metabolism, excretion and toxicity within living organisms [59]. ADMET predictions of the synthesised 1,2,3-triazole–quinoline–aniline compounds were calculated using the QikProp utility [60] in the Schrödinger Suite [49] (Table 5). The human serum binding ability coefficient (QPlogKhsa) ranged from 0.500 to 0.779, suggesting a likely favourable bioavailability and their being less likely to be protein-bound. The predicted aqueous solubility (QPlogS) values for all but one compound, **11g** (−6.979), were within the acceptable range, suggesting good intestinal absorption.

The predicted percentage of human oral absorption for most compounds was 100%, except for **11i**, which had ~88%, indicating a potential excellent oral bioavailability, with >80% being categorised as high. Furthermore, the predicted brain/blood coefficient (QPlogBB) values were also in the accepted range (−0.023 to −1.346), while these compounds were predicted to be likely inactive in the central nervous system (CNS) (<+2). The likely number of metabolic reactions from the cytochrome P450 enzyme was predicted to be less than seven, indicating a favourable outcome. Lastly, most of these compounds adhered to Lipinski’s criteria for molecular weight, the octanol–water coefficient, and the number of hydrogen bond donors and acceptors, except **11c** and **11g**, which violated this guideline due to their high lipophilicity (clogP) of greater than 5.

## 3. Materials and Methods

### 3.1. Chemistry

All the chemical reagents used in the synthesis were purchased from Merck South Africa/Sigma Aldrich (Modderfontein, South Africa) and I&A Chemicals (Namyangju, Republic of Korea), with a purity ranging from 97 to 100%. HPLC-grade and crude solvents were used. The reaction progress was monitored using thin-layer chromatography (TLC) analysis on aluminium-backed TLC plates (Kiese gel 60 F254 plates, Merck South Africa) and visualised under ultraviolet light (254 nm wavelength) All the synthesised final compounds and some intermediates were purified using flash-column chromatography on silica gel (0.063–0.200 mm) and various solvent systems. A melting point analysis was conducted using an electrothermal IA9100 melting point apparatus (Rochford, Essex, UK) on the solid compounds using glass capillary tubes; the melting points are recorded in degrees celsius (℃) and are uncorrected. The final compounds’ and intermediates’ functional groups were analysed and confirmed by Fourier-transform infrared (FTIR) spectroscopy on the Perkin Elmer 100 spectrophotometer (Waltham, MA, USA) with the Universal ATR sampling accessory; wavenumbers (υ) on the spectra are expressed in cm^−1^.

Nuclear magnetic resonance (NMR) analysis was conducted on the Bruker Avance III 600 Hz spectrometer (Billerica, MA, USA) using deuterated chloroform (CDCl_3_) and dimethyl sulfoxide (DMSO-*d*_6_) and solvents. Topspin was used for the spectra analysis; the coupling constants (*J*) are reported in Hertz (Hz) and the chemical shifts in parts per million (ppm) using the tetramethyl silane (TMS) peak as a reference. The splitting patterns are reported as singlet (s), doublet (d), multiplet (m), triplet (t), quartet (q), doublet of doublets (dd), doublet of triplets (dt) or triplet of doublets (td). The solvent peaks were referenced at 2.50 (^1^H) and δ 39.5 (^13^C) for DMSO-*d*_6_, and 7.26 (^1^H) and 77.0 (^13^C) for CDCl_3_, while residual water was observed at 3.35 and 1.56 ppm, respectively.


**Preparation of 4-azido-7-chloroquinoline (7)**


This compound was prepared as previously reported by Nyoni et al. [22]. Briefly, a mixture of 4,7-dichloroquinoline (2.0 g, 10 mmol), molecular sieves A4, and sodium azide (1.3 g, 20 mmol) in 5 mL anhydrous DMF was refluxed at 85 °C for 24 h. The reaction was monitored by TLC, and the mixture was cooled to room temperature upon completion. The cooled mixture was diluted with 100 mL DCM and washed with brine solution (3 × 40 mL). The organic extract was dried over anhydrous sodium sulphate, filtered, and concentrated in vacuo. The crude product obtained was then subjected to column chromatography on silica gel, using a mobile phase of DCM/hexane (1:1) to afford 4-azido-7-chloroquinoline (2) as off-white needle-like crystals; 1.76 g (85%), mp: 113–115 °C, IR (cm^−1^) 2118 (N_3_). ^1^H-NMR (600 MHz, DMSO-*d*_6_, ppm): δ_H_ 7.25 (1H, d, *J* = 2.01 Hz, H-3), 7.43 (1H, dd, *J*_1_ = 1.76, *J*_2_ = 8.21, H-6), 7.83 (1H, d, *J* =1.68 Hz, H-8), 7.93 (1H, d, *J* = 8.96 Hz, H-5), 8.66 (1H, d, *J* = 5.0 Hz, H-2).

**Preparation of the propargylated** (**alkyne**) **compounds** (**10a–j**)**. Compound 10a is chosen as representative.**

Propargylated benzylamines were synthesised following the method reported by Mao et al. [43]. Briefly, 198 µL (2.6 mmol) of 70% propargyl bromide (**9**) was added dropwise, while stirring, to a 50 mL round-bottom flask containing 3.92 mmol of the respective aniline and dry DCM. Thereafter, 361 mg of potassium carbonate (K_2_CO_3_), activated at 300 °C, was then added to the reaction mixture. The resultant reaction mixture was allowed to be stirred at room temperature for 24 h. Upon completion of the reaction, DCM was evaporated in vacuo, and the resulting residues were further dried under a vacuum for 24 h. The crude products were purified by column chromatography to obtain alkynes **10a**–**j**. Compound **5a** is used as a representative compound below.

*N-2-propyl-1-yl-benzanamine* (**10a**) (reference) as a brown liquid, 132 mg (43%) yield, ^1^H-NMR (DMSO-*d*_6_, 600 MHz, ppm) δ_H_ 3.0 (1H, t, *J* = 2.4 Hz, H-6), 3.8 (2H, dd, *J*_1_ = 6.1; *J*_2_ = 2.40 Hz, H-1), 5.9 (1H, t, *J* = 6.1 Hz, H-2), 6.6 (1H, t, *J* = 7.04 Hz, H-5), 6.63 (2H, d, *J* = 8.14 Hz, H-3), 7.1 (2H, t, *J* = 8.4 Hz, H-4). Compound **10a** is used as a representative compound below.

Alkynes **10b**–**j** were characterised and confirmed in a similarly manner and corresponded with the published data for each [41,42,43,44].

**Preparation of *N*-[1-**(**7-chloroquinolin-4-yl**)**-1*H*-1,2,3-triazol-4-yl]anilines** (**11a–j**)**.**

An amount of 100 mg (0.77 mmol) of the respective *N*-(prop-2-ny-l-yl)anilines (**10a**–**j**) and 189 mg (0.92 mmol) 7-chloroquinoline-4-azide (**7**) was dissolved in 10 mL DCM in a 100 mL round-bottom flask. Thereafter, 108.6 mg (22%) sodium ascorbate, 39.9 (10%) copper sulphate and 10 mL of water was added, and the reaction mixture was vigorously stirred at room temperature until completion (24 h). On completion, based on TLC, 100 mL of water was added, followed by 5 × 40 mL DCM extraction, and the combined extracts evaporated in vacuo to afford crude compounds, which were purified by column chromatography (DCM: MeOH; 95:5%):*1-(7-Chloro-4-quinolinyl)-1H-1,2,3,-triazole-4-methanamine* (**11a**): As a cream-white solid, yield 228.4 mg (88%), mp 15–152 °C, IR (cm^−1^) C-H 2900–3000, ^1^H-NMR (DMSO-*d*_6_, 600 MHz, ppm) δ_H_ 4.47 (2H, d, *J* = 5.7 Hz, H-1), 6.21 (1H, t, *J* = 5.7 Hz, H-2), 6.58 (1H, t, *J* = 7.5 Hz, H-5), 6.72 (2H, d, *J* = 7.9 Hz, H-3 and H-3′), 7.11 (2H, dd, *J*_1_ = 8.3 Hz, *J*_2_ = 7.4 Hz, H-4 and H-4′), 7.78 (1H, dd, *J*_1_ = 9.0 Hz, *J*_2_ = 2.0 Hz, H-10), 7.8 (1H, d, *J* = 4.5 Hz, H-7), 8.02 (1H, d, *J* = 9.1 Hz, H-11), 8.28 (1H, d, *J* = 2.0 Hz, H-9), 8.74 (1H, s, H-6), 9.14 (1H, d, *J* = 4.5 Hz, H-8). ^13^C-NMR (DMSO-*d*_6_, 150 MHz, ppm) δ_C_ 39.9 (C-1), 112.9 (C-3), 116.7 (C-5), 117.3 (C-7), 120.7 (C-15a), 125.7 (C-11), 125.9 (C-6), 128.6 (C-9), 129.4 (C-4,10), 135.8 (C-16), 140.9 (C-14), 147.1 (C-13), 148.7 (C-12), 149.9 (C-15b), 152.8 (C-8). TOFF MS ES^−^: (*m*/*z*) 306.0968 (100%) [(M − H) − N_2_]^−^ (Calculated for C_18_H_13_ClN_3_^−^ (306.0803)].*[1-(7-Chloro-4-quinolinyl)-1H-1,2,3-triazole-4-methyl]-4-bromoaniline* (**11b**): As a light brown solid, yield 278.2 mg (87%), mp 187–190 °C, IR (cm^−1^) C-H 2900–3000; ^1^H-NMR (DMSO-*d*_6_, 600 MHz, ppm) δ_H_ 4.48 (2H, d, *J* = 5.2 Hz, H-1), 6.4 (1H, s, H-2), 6.71 (2H, d, *J* = 8.7 Hz, H-4), 7.25 (2H, d, *J* = 8.7 Hz, H-3), 7.76 (1H, dd, *J*_1_ = 9.0, *J*_2_ = 1.5 Hz, H-10), 7.81 (1H, d, *J* = 4.6 Hz, H-7), 8.01 (1H, d, *J* = 9.0 Hz, H-11), 8.26 (1H, d, *J* = 1.5 Hz, H-9), 8.74 (1H, s, H-6), 9.14 (1H, d, *J* = 4.5 Hz, H-8). ^13^C-NMR (DMSO-*d*_6_, 150 MHz, ppm) δ_C_ 38.0 (C-1)114.0 (C-4), 131.0 (C-3), 117.3 (C-7), 120.7 (C-15a), 125.6 (C-6), 125.8 (C-11), 128.6 (C-9), 129.4 (C-10), 135.8 (C-16), 140.9 (C-14), 148.0 (C-12), 147.0 (C-13), 149.8 (C-15b), 152.0 (C-8), 107.0 (C-5). TOF MSMS ES^−^: (*m*/*z*) 451.4776 [(M^−^ + HCl]^−^ [Calculated for C_18_H_19_BrCl_2_N_5_ (451.1490)].*[1-(7-Chloro-4-quinolinyl)-1H-1,2,3-triazole-4-methyl]-4-iodoaniline* (**11c**): As a brown solid, yield 328.1 mg (92%), mp 195–199 °C, IR (cm^−1^) C-H 2900–3000; ^1^H-NMR (DMSO-*d*_6_, 600 MHz, ppm) δ_H_ 4.45 (2H, d, *J* = 5.5 Hz, H-1), 6.47 (1H, t, *J* = 5.5 Hz, H-2), 6.58 (2H, d, *J* = 8.7 Hz, H-4), 7.37 (2H, d, *J* = 8.7 Hz, H-3), 7.78 (1H, dd, *J*_1_ = 9.0, *J*_2_ = 1.9 Hz, H-10), 7.81 (1H, d, *J* = 4.5 Hz, H-7), 7.99 (1H, d, *J* = 8.8 Hz, H-11), 8.28 (1H, s, H-9), 8.72 (1H, s, H-6), 9.15 (1H, s, H-8). ^13^C-NMR (DMSO-*d*_6_, 150 MHz, ppm) δ_C_ 39.0 (C-1), 137.6 (C-3),117.3 (C-7), 77.4 (C-5), 120.7 (C-15a), 125.6 (C-6), 125.9 (C-11), 128.5 (C-9), 129.3 (C-10), 115.8 (C-4), 135.7 (C-16), 140.9 (C-14), 137.9 (C-12), 147.1 (C-13), 149.8 (C-15b), 152.8 (C-8). TOFF MS ES^−^: (*m*/*z*) [(M + Cl]^−^ 495.9821 (100%) (Calculated for C_18_H_13_Cl_2_N_5_^−^ (495.9593)].*[1-(7-Chloro-4-quinolinyl)-1H-1,2,3-triazole-4-methyl]-4-flouroaniline* (**11d**): As a light grey solid, yield 216.0 mg (79%), mp-145–148, IR (cm^−1^) C-H 2900–3000, ^1^H-NMR (DMSO-*d*_6_, 600 MHz, ppm) δ_H_ 4.44 (2H, d, *J* = 5.2 Hz, H-1), 6.15 (1H, t, *J* = 5.6 Hz, H-2), 6.71 (2H, dd, *J*_1_ = 9.1 Hz, ^2^*J*_(H-F)_ = 4.4 Hz, H-4), 6.96 (2H, t, *J* = 9.1 Hz, H-3), 7.79 (1H, dd, *J*_1_ = 9.0 Hz, ^3^*J*_(H-F)_ = 1.5 Hz, H-10), 7.82 (1H, d, *J* = 4.5 Hz, H-7), 8.01 (1H, d, *J* = 9.0, H-11), 8.29 (1H, d, *J* = 1.5 Hz, H-9), 8.73 (1H, s, H-6), 9.15 (1H, d, *J* = 4.5 Hz, H-8). ^13^C-NMR (DMSO-*d*_6_, 150 MHz, ppm) δ_C_ 39.0 (C-1), 113.7 (d, *^3^J*_(C-F)_ = 7.02 Hz, C-3), 115,8 (d, *^2^J*_(C-F)_ = 21.9 Hz, C-4), 117.3 (C-7), 120.7 (C-15a), 125.6 (C-6), 125.8 (C-11), 128.6 (C-9), 129.4 (C-10), 135.8 (C-16), 140.9 (C-14), 145.4 (C-12), 147.0 (C-13), 149.8 (C-15b), 152 (C-8), 154.3/155.8 (d, ^1^*J_(_*_C-F)_ = 230.04 Hz, C-5). TOFF MS ES^−^: (*m*/*z*) [(M − H) − N_2_]^−^ 324.0866 (100%) (Calculated for C_18_H_12_ClFN_3_^−^ (324.0709)].*[1-(7-Chloro-4-quinolinyl)-1H-1,2,3-triazole-4-methyl]-3-chloroaniline* (**11e**): As a yellow solid, yield 197.2 mg (69%), mp 169–172 °C, IR (cm^−1^) C-H 2900–3000, ^1^H-NMR (DMSO-*d*_6_, 600 MHz, ppm) δ_H_ 4.48 (2H, d, *J* = 5.6 Hz, H-1), 6.53 (1H, t, *J* = 9.2 Hz, H-2), 6.66 (H, t, *J* = 8.0 Hz, H-5), 6.74 (1H, s, H-3′), 6.74 (1H, dd, *J*_1_ = 3 Hz, *J*_2_ = 2.4 Hz H-4′), 7.10 (1H, dd, *J*_1_ = 9.2 Hz, *J*_2_ = 1.9 Hz, H-3), 7.12 (1H, s, H-3), 7.80–7.79 (2H, m, H-7 and H-10), 8.01 (1H, d, *J* = 9.0 Hz, H-11), 8.27 (1H, d, *J* = 1.6 Hz, H-9), 8.72 (1H, s, H-6), 9.13 (1H, d, *J* = 4.5 Hz, H-8). ^13^C-NMR (DMSO-*d*_6_, 150 MHz, ppm) δ_C_ 38.6 (C-1), 112.1 (C-3), 111.5(C-3′), 117.3 (C-7), 130.8 (C-5), 120.7 (C-15a), 125.7 (C-6), 125.8 (C-11), 128.6 (C-9), 129.3 (C-10), 124.7 (C-4′), 135.8 (C-16), 134.1 (C-4), 140.9 (C-14), 150.2 (C-12), 147.0 (C-13), 149.8 (C-15b), 152.8 (C-8). TOFF MS ES^−^: (*m*/*z*) [(M − H) − N_2_]^−^ 340.0580 (100%) (Calculated for C_18_H_12_Cl_2_N_3_^−^ (340.0414)].*[1-(7-Chloro-4-quinolinyl)-1H-1,2,3-triazole-4-methyl]-2-methoxyaniline* (**11f**): As a brown liquid, yield 121.6 mg (43%), IR (cm^−1^) C-H 2900–3000 ^1^H-NMR (DMSO-*d*_6_, 600 MHz, ppm) δ_H_ 3.79 (3H, s, H-3″), 4.52 (2H, d, *J* = 6 Hz, H-1), 5.46 (1H, t, *J* = 6 Hz, H-2), 6.59 (1H, td, *J*_1_ = 7.7 Hz, *J*_2_ = 1.4 Hz, H-5), 6.72 (1H, dd, *J*_1_ = 7.7 Hz, *J*_2_ = 1.4, H-3′), 6.78 (1H, d, *J* = 8.0 Hz, H-4), 6.83 (1H, td, *J*_1_ = 7.7 Hz, *J*_2_ = 0.6 Hz, H-4′), 7.76 (1H, dd, *J*_1_ = 9.1 Hz, *J*_2_ = 2.1 Hz, H-10), 7.81 (1H, d, *J* = 4.6 Hz, H-7), 8.01 (1H, d, *J* = 9.1 Hz, H-11), 8.26 (1H, d, *J* = 2.0 Hz, H-9), 8.69 (1H, s, H-6), 9.12 (1H, d, *J* = 4.6 Hz, H-8). ^13^C-NMR (DMSO-*d*_6_, 150 MHz, ppm) δ_C_ 39.0 (C-1), 110.2 (C-3′), 138.0 (C-3), 117.3 (C-7), 116.7 (C-5), 120.7 (C-15a), 125.6 (C-6), 125.9 (C-11), 128.5 (C-9), 129.3 (C-10), 110.3 (C-4), 135.7 (C-16), 121.4 (C-4′), 140.9 (C-14), 137.9 (C-12), 147.1 (C-13), 149.8 (C-15b), 152.8 (C-8), 55.7 (C-3″). TOF MSMS ES^+^: (*m*/*z*) 388.1096 (M + Na)^+^ [Calculated for C_19_H_16_ClN_5_NaO (388.0941)].*[1-(7-Chloro-4-quinolinyl)-1H-1,2,3-triazole-4-methyl]-2-triflouromethylaniline* (**11g**): As a yellow liquid, yield 115.3 mg (48%), IR (cm^−1^) C-H 2900–3000 ^1^H-NMR(DMSO-*d*_6_, 600 MHz, ppm) δ_H_ 4.65 (2H, d, *J* = 5.7 Hz, H-1), 6.11 (1H, s, H-2), 6.73 (1H, t, *J* = 7.5 Hz, H-5), 7.01 (1H, d, *J* = 8.4 Hz, H-3′), 7.43 (1H, m, H-4,4′), 7.76 (1H, dd, *J*_1_ = 9.0 Hz, *J*_2_ = 2.0 Hz, H-10), 7.80 (1H, d, *J* = 4.6 Hz, H-7), 7.96 (1H, d, *J* = 9.0 Hz, H-11), 8.25 (1H, d, *J* = 2.0, H-9), 8.65 (1H, s, H-6), 9.11 (1H, d, *J* = 4.6 Hz, H-8). ^13^C-NMR (DMSO-*d*_6_, 150 MHz, ppm) δ_C_ 39.0 (C-1), 145.4 (C-3), 112.9 (C-3″), 117.4 (C-7), 116.2 (C-5), 120.7 (C-15a), 126.1 (C-6), 125.8 (C-11), 128.5 (C-9), 129.3 (C-10), 126.7 (C-4′), 135.8 (C-16), 134.0 (C-4), 140.9 (C-14), 146.5 (C-12), 146.5 (C-13), 149.8 (C-15b), 152.8 (C-8), 126 (C-3′). LCMS (ACN with 0.1% formic acid) (*m*/*z*) (M + 1) 404.*[1-(7-Chloro-4-quinolinyl)-1H-1,2,3-triazole-4-methyl]-3-flouroaniline* (**11h**): As a cream-white solid, yield 149.7 mg (85%), mp 145–148 °C, IR (cm^−1^) C-H 2900 ^1^H-NMR (DMSO-*d*_6_, 600 MHz, ppm) δH 4.47 (2H, d, *J* = 5.7 Hz, H-1), 5.40 (1H, s, H-2), 6.35 (1H, td, *J*_1_ = 8.6, *J*_2_ = 2 Hz, H-4), 6.49–6.55 (3H, m, H-3, H-3′ and H5), 7.76 (1H, dd, *J*_1_ = 9.0 Hz, *J*_2_ = 1.9 Hz, H-10), 7.80 (1H, d, *J* = 4.7 Hz, H-7), 8.00 (1H, d, *J* = 9.2 Hz, H-11), 8.28 (1H, s, Hz, H-9), 8.72 1(H, s, H-6), 9.13 (1H, d, *J* = 4.6 Hz, H-8). ^13^C-NMR (DMSO-*d*_6_, 150 MHz, ppm) δ_C_ 38.7 (C-1), 99.2 (d, ^2^*J*_C-_F = 33.2 Hz, C-3), 109.2 (C-3′), 117.3 (C-7), 130.7 (d, ^2^*J*_C-F_ = 30.1 Hz, C-5), 143.5 (d, ^3^*J*_C-F_ = 4.1 Hz, C-4′) 120.7 (C-15a), 125.6 (C-6), 125.9 (C-11), 128.5 (C-9), 129.4 (C-10), 135.8 (C-16), 165.1(C-4), 140.9 (C-14), 153.5 (C-12), 146.6 (C-13), 149.8 (C-15b), 152.8 (C-8). TOFF MS ES^−^: (*m*/*z*) [(M − H) − N_2_]^−^ 324.0868 (100%) (Calculated for C_18_H_12_ClFN_3_^−^ (324.0709)]. *[1-(7-Chloro-4-quinolinyl)-1H-1,2,3-triazole-4-methyl]-3-nitroaniline* (**11i**): As an orange solid, yield 232.4 mg (79%), mp 192–197 °C, IR (cm^−1^) C-H 2900–3000. ^1^H-NMR(DMSO-*d*_6_, 600 MHz, ppm) δ_H_ 4.58 (2H, d, *J* = 5.7 Hz, H-1), 7.00 (H, s, H-2), 7.15 (1H, d, *J* = 7.74 Hz, H-3′), 7.37 (2H, m, H-4′ and H-5), 7.52 (1H, d, *J* = 7.7 Hz, H-3), 7.77 (1H, t, *J* = 2.0 Hz, H-10), 7.78 (1H, dd, *J*_1_ = 8.7 Hz, *J*_2_ = 2, H-7), 7.81 (H, dd, *J*_1_ = 9.0 Hz, *J*_2_ = 1.5 Hz, H-11), 8.29 (1H, d, *J* = 2 Hz, H-9), 8.77 (1H, s, H-6), 9.14 (1H, d, *J* = 4.5 Hz, H-8). ^13^C-NMR (DMSO-*d*_6_, 150 MHz, ppm) δ_C_ 39.1 (C-1), 130 (C-4′), 106 (C-3′), 117.4 (C-7), 119.1 (C-3) 120.8 (C-15a), 125.8 (C-6), 125.8 (C-11), 128.6 (C-9), 129.4 (C-10), 135.8 (C-16), 140.9 (C-14), 145.4 (C-12), 146.2 (C-13), 149.81 (C-4), 149.8 (C-15b), 152 (C-8), 111.0 (C-5). TOF MSMS ES^+^: (*m*/*z*) 381.2533 (M + 1)^−^ [Calculated for C_18_H_14_ClN_6_O_2_ (381.7920)].*[1-(7-Chloro-4-quinolinyl)-1H-1,2,3-triazole-4-methyl]-4-methoxyaniline* (**11j**): As a brown solid, yield 243.1 mg (86%), mp 169–172, IR (cm^−1^) C-H 2900–3000. ^1^H-NMR(DMSO-*d*_6_, 600 MHz, ppm) δ_H_ 3.64 (3H, s, H-5′), 4.42 (2H, d, *J* = 5.8 Hz, H-1), 5.76 (1H, t, *J* = 5.8 Hz, H-2), 6.68 (2H, d, *J* = 8.8 Hz, H-4), 6.74 (2H, d, *J* = 8.8 Hz, H-3), 7.75 (1H, dd, *J*_1_ = 9.0 Hz, *J*_2_ = 1.8 Hz, H-10), 7.79 (1H, d, *J* = 4.6 Hz, H-7), 8.01 (1H, d, *J* = 9.0 Hz, H-11), 8.25 (1H, d, *J* = 1.7 Hz, H-9), 8.68 (1H, s, H-6), 9.11 (1H, d, *J* = 4.6 Hz, H-8). ^13^C-NMR (DMSO-*d*_6_, 150 MHz, ppm) δ_C_ 40.4 (C-1), 115.07 (C-3), 117.2 (C-7), 151.6 (C-5), 55.7 (C-5′), 120.7 (C-15a), 125.6 (C-6), 125.9 (C-11), 128.5 (C-9), 129.3 (C-10), 135.7 (C-16), 114.1 (C-4), 140.9 (C-14), 142.9 (C-12), 147.1 (C-13), 149.8 (C-15b), 152.8 (C-8). TOFF MS ES^−^: (*m*/*z*) [(M − H) − N_2_]^−^ 336.1097 (100%) [Calculated for C_19_H_15_ClN_3_O^−^ (336.0909)].

### 3.2. Biology

#### 3.2.1. Antimycobacterial Evaluation

An in vitro antimycobacterial evaluation assay was performed against the H37Rv strain using the previously reported procedure [22]. Cultured H37Rv (ATCC 27294) in Middlebrook 7H9 (Difco, Becton Dickinson, Franklin Lakes, NJ, USA) broth supplemented with 0.1% glycerol (Merck, Darmstadt, Germany) and 10% oleic acid–albumin–dextrose–catalase (OADC) (Becton-Dickenson) was aerobically grown at 37 °C until an optical density (OD)600 nm of 1 was attained. This was equivalent to approximately 3 × 108 bacilli/mL.

The antimicrobial activity of the various compounds was tested in triplicate using micro broth dilution assays in 96-well plates. These plates were sealed and incubated at 37 °C for 7 days, and microbial growth was measured by observing the resazurin colour change from blue to pink. The minimum inhibitory concentration (MIC) was interpreted as the lowest concentration inhibiting a colour change from blue to pink.

#### 3.2.2. MTT Cytotoxicity Evaluation

An in vitro cytotoxicity evaluation assay was performed on the TZM-bl cell line, a HeLa cell line clone, as previously reported [22]. The cells were seeded at a density of 25,000 (DEAE dextran 44 µL/10 mL) cells + 150 µL DMEM/well in a 96-well microtiter plate in duplicates and incubated overnight for attachment (37 °C, 5% CO_2_). Treatments (2.5 mg/mL) were prepared, and following incubation, the supernatant (treatment medium) was removed, and 120 µL of MTT solution comprising 100 µL fresh CCM and 20 µL of MTT (5 mg/mL MTT salt in 0.1 M PBS) was added to each well. The plate was then incubated for 4 h (37 °C, 5% CO_2_). The optical density of each sample was measured at 450 using a microplate reader (Perkin Elmer, Waltham, MA, USA). The maximum inhibitory concentration resulting in a 50% cytotoxicity concentration (CC_50_) was obtained using GraphPad Prism version 5.01 by plotting a dose–response curve (concentration versus the percentage cell viability of the samples) (see cytotoxicity dose–response curve in the Appendix A).

#### 3.2.3. Luciferase-Based Antiviral Assay Evaluating Human Immunodeficiency Virus


*Maintenance of cell lines*


In sterile 75 cm^2^ culture flasks, the TZM-bl cell lines (NIH AIDS Research and Reference Reagents Programme) were cultured as a monolayer using Dulbecco’s Modified Eagle Medium (DMEM) (Thermo Fisher Scientific, Waltham, MA, USA) supplemented with 10% foetal bovine serum (FBS; heat-inactivated and gamma-irradiated) (LTC Biosciences, Gainesville, FL, USA), 25 mM of HEPES (Thermo Fisher Scientific, Waltham, MA, USA) and 50 μL/mL of gentamicin (Thermo Fisher Scientific, Waltham, MA, USA). A HeLa cell line clone, the TZM-bl cell line is altered to produce CD4 and CCR5, enabling HIV-1 infection and firefly luciferase regulated by the HIV-1 long-terminal repeat (LTR) [61].


*Antiviral*
*assay*


The HIV-1 inhibition of the synthesised drugs was evaluated using a luciferase-based antiviral assay [62]. Initially, 96-well cell culture plates were filled with 150 uL, 100 uL and 140 uL of DMEM; the test compound, viral control and cell control were added, respectively. Briefly, the 96-well cell culture plates (Corning Costar, New York, NY, USA) were filled with 11 uL of the AZT drug (positive control) and test compounds. The plates were then diluted three times in 140 uL of DMEM supplemented with 10% FBS, 25% HEPES buffer and 1% penicillin-streptomycin. A total of 10,000 TZM-bl cell lines were infected with 50 uL of NL4.3 virus (subtype B) in the 96-well culture plates. The experimental controls included the infected (virus control) and uninfected (cell control) TZM-bl cell line, which was incubated for an hour. After adding 10,000 cells to each 96-well plate, the cells were grown for 48 h at 37 °C, 5% CO_2_, 95% humidity, and with 37.5 ug/mL of DEAE–dextran. A total of 150 uL of medium was taken out and replaced with 100 uL of the Bright-Glo TM luciferase reagent without light exposure following a 48 h incubation period. After aspirating the supernatant, 150 uL of the mixture containing the Bright-Glo TM luciferase reagent was put into a Corning Costar 96-well black plate. It was measured right away at 540 nm in a Victor Nivo microplate reader (PerkinElmer, Waltham, MA, USA). Then, the percentage of viral inhibition was calculated as follows:% HIV inhibition = (average sample − average control)/(1 − (average viral control − average control) × 100

The results of the absorbance-based quantification of the viral cell through the inhibitory concentration at 50% was obtained by plotting the dose–response curve (log concentration versus % HIV inhibition) (see HIV assay dose–response curved in the Appendix A).

### 3.3. In Silico Studies

#### 3.3.1. Molecular Docking

The compounds and proteins were prepared using the ligand preparation (LigPrep) and protein preparation wizard modules [63] (on Maestro software 13.9 in the Schrödinger Suite [49]). The compounds were docked at the active sites of the proteins.

#### 3.3.2. DFT Studies

The structure of 11 h was optimised at the DFT/B3LYP/6-311++G(d,p) level of theory in the gas phase. Frequency calculations of the optimised structure were performed to ensure that the geometry conformed to minima. The ionisation potential and electron affinity were calculated from the frontier molecular orbital energies (EHOMO) and (ELUMO), respectively [64]. Other reactivity indices such as the energy gap (Eg), chemical hardness and softness (η and S, respectively), electronegativity (χ) and electrophilicity (ω) were all calculated [65]; see Equations (1)–(8). The distribution of the molecular orbitals over the molecular surface was visualised via the HOMO and LUMO maps [58]. An electrostatic potential (ESP) map was used to visualise the selective reactive sites of interaction of 11 h with an electron-donating or -withdrawing neighbour [66].(1)$$I={-E}_{HOMO}$$(2)$$A={-E}_{LUMO}$$(3)$$E_{g}=E_{LUMO}-E_{HOMO}$$(4)$$\eta =\frac{I-A}{2}$$(5)$$S=\frac{1}{\eta }$$(6)$$\chi =\frac{I+A}{2}$$(7)$$\omega =\frac{{\left(I+A\right)}^{2}}{8\eta }=\frac{\chi^{2}}{2\eta }$$

## 4. Conclusions

Applying molecular hybridisation, new quinoline–1,2,3-triazole–anilines were successfully synthesised in moderate to excellent yields. Their structures were confirmed using spectroscopic and spectrometric techniques. The synthesised compounds demonstrated moderate to negligible activity against *Mtb* in vitro. However, notable anti-HIV activity was observed in compounds **11g**, **11h** and **11i**, with their IC_50_ being 0.3883 µM, 0.0103 µM and 0.167 µM, respectively, with **11h** exhibiting the best activity against both *Mtb* and HIV. Furthermore, **11h** showed a 9-fold superior activity than the reference drug, AZT (0.0909 µM).

Additionally, the presence of fluoride in certain compounds appears to have improved their antiviral activity. Cytotoxicity assessments generally revealed low toxicity, except for a few compounds. The selective indices (SIs) of **11g** and **11h** are 11,367 and 2472.87, respectively, suggesting that these compounds would pose less cytotoxic effects in vivo. Molecular docking studies revealed that **11h** interacted with some of the essential amino acid residues in the active site of the HIV-1 co-receptor entry enzyme. The DFT studies on **11h** revealed reactivity and reactive sites in the compound, while the predicted ADMET parameters for most compounds indicated drug-like molecules. Thus, **11h** is a potential hit for further optimisation studies against HIV-1.

## Data Availability

Original experimental data not provided in the Appendix A are available from the authors on request.

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
