# Peer review of "Synthesis, Characterisation, Biological Evaluation and In Silico Studies of Quinoline–1,2,3-Triazole–Anilines as Potential Antitubercular and Anti-HIV Agents"

_molecules, 2025, doi:10.3390/molecules30102119_

Round 1
Reviewer 1 Report
Comments and Suggestions for Authors
Dear Author
Snethemba S. Magwaza and co-workers discussed about quinoline-1,2,3-triazole-anilines as potential antitubercular and anti-HIV agents. Spectral methods characterize the developed compounds. Prepared compounds with in vitro activities against the wild-type HIV-1 subtype B as well as against Mtb’s H37Rv strain, are interesting, which means these compounds are Dual-targeted anti-TB/anti-HIV. The manuscript is written very smoothly and is easy to understand for the reader. The Manuscript is suitable for publication, but with Minor revisions.
- Materials and Methods
Correct DMSO-d6, it always DMSO-d6
J should be J (Italic), this correction all over the 11a to 11J spectral data
- In what region have you made modifications to the phenyl ring? There was no functional modification on the quinoline ring.
- Why are you reporting two dual activities in one manuscript?
With this correction. The manuscript is suitable for acceptance
Thanks & regards
Author Response
Please cover letter

Reviewer 2 Report
Comments and Suggestions for Authors
The search for new drugs to treat tuberculosis and HIV is an essential and urgent challenge. In this context, the article by Matshawandile Tukulula et al., focused on developing anti-TB and anti-HIV agents, is scientifically sound and relevant. However, before it can be published, the authors need to address several comments, which are listed below.
- It is necessary to provide a decoding of the substituents R, R1, R2, and R3 in Scheme 1.
- Comments are needed regarding compounds 7 and 10a presented in Table 2.
- There are numerous targets (active sites) for docking anti-TB agents. Why did the authors select the ATP synthase enzyme (PDB ID: 4VIF) without performing a reverse docking strategy (see, for example, https://doi.org/10.3390/ijms26010369)?
- It is essential to specify the units of measurement for the "docking scores" listed in Table 3.
- The authors should provide a more detailed explanation of how the descriptors in Table 4 relate to anti-TB and anti-HIV activities.
- The authors should provide a more detailed explanation of how the section "HOMO, LUMO, and ESP surface maps of 11h" relates to biological activity. Are there any literature data on the correlation between the values of HOMO, LUMO, and ESP surface maps and antibacterial or antiviral activities?
- Are there any literary sources that explain which values of QPlogKhsa, QPlogS, and QPlogBB provided in Table 5 are considered promising or not? Are there critical values for these quantities?
- In section "3. Materials and Methods," the authors must provide a detailed description of the synthesis and analytical data for all intermediates 7 and 10, including all spectral data in the Supporting Information.
- 22% sodium ascorbate, 10% copper sulphate. Are these mole percent? You also need to give the quantities in mg.
- All product yields should be given in mg.
- It is essential to provide 19F NMR spectra for the fluorine-containing target products 11d, 11g, and 11h, as well as for the intermediates 10d, 10g, and 10h.
- What is the reason for the absence of high-resolution mass spectrometry data for compound 11g?
Round 2
Reviewer 2 Report
Comments and Suggestions for Authors
Authors have carefully checked and modified this manuscript. Now it can be accepted for publication in this journal without further revision.